# Causes and trends in liver disease and hepatocellular carcinoma among men and women who received liver transplants in the U.S., 2010-2019

**Sonia Wang, Mehlika Toy, Thi T. Hang Pham[ORCID], Samuel So***

Asian Liver Center, Department of Surgery, Stanford University School of Medicine, Stanford, California, United States of America

* samso@stanford.edu

## Abstract

### Background and aims

The national Organ Procurement and Transplant Network (OPTN) reported the major indication for liver transplants in 2018 was for other/unknown causes. This study was undertaken to examine all causes and trends in liver disease and hepatocellular carcinoma (HCC) among adults who received liver transplants in the past 10 years.

### Methods

A national cohort study of all adults who received liver transplants from Jan 1, 2010 to Dec 31, 2019 recorded in the OPTN STAR database analyzed by etiology of liver disease and HCC, and gender.

### Results

Adult liver transplants increased from 5,731 in 2010 to 8,345 in 2019 (45.6% increase). Between 2010 and 2014, liver disease and HCC associated with hepatitis C (HCV) was the major cause for liver transplantation. Proportion of liver transplants for HCV associated liver disease and HCC has since decreased to 18.7% in 2019 compared with 44.5% in 2010 [25.8%, (95% CI 24.3% to 27.3%), p<0.001], while liver transplants for liver disease and HCC associated with alcohol-associated liver disease (ALD) and non-alcoholic fatty liver disease (NAFLD) increased from 12.7% to 28.8% [16.1%, (95% CI 14.8% to 17.4%), p<0.001], and from 9.1% to 21.5% [12.4%, (95% CI 11.2% to 13.5%), p<0.001], respectively. When all causes of liver disease were examined, only 1.7% of liver transplants had unspecified causes. The five major causes of liver disease and HCC among men receiving liver transplants in 2019 were ALD (33.1%), HCV (21.9%), NAFLD (18.5%), cholestatic liver disease (5.7%) and hepatitis B (4.9%), while the major causes among women were NAFLD (26.8%), ALD (21.1%), HCV (13.1%), cholestatic liver disease (11.1%), and autoimmune liver disease (5.6%).

**Data Availability Statement:** All relevant data are within the manuscript and its Supporting Information files.

**Funding:** All the funding and support for this study is provided by the Asian Liver Center, Department of Surgery, Stanford University School of Medicine. There was no additional external funding received for this study.

**Competing interests:** The authors have declared that no competing interests exist

**Abbreviations:** ALD, alcohol associated liver disease (non-HCV related); BMI, body mass index; HBV, hepatitis B; HCC, hepatocellular carcinoma; HCV, hepatitis C; LT, liver transplant; NAFLD, non-alcoholic fatty liver disease; OPTN, Organ Procurement and Transplant Network.

## Conclusions

Our study found NAFLD in 2017 in women and ALD in 2019 in men have surpassed HCV as the leading causes of liver disease and HCC among adults receiving liver transplants.

## Introduction

Although end stage liver disease and HCC caused by chronic hepatitis C have long been the leading indications for liver transplantation in adults in the U.S. [1], recent studies highlighted the increasing number of people with non-alcoholic fatty liver disease (NAFLD) and non-HCV related alcohol associated liver disease (ALD) added onto the U.S. liver transplant waiting list [2–5]. This trend has been attributed to the introduction of curative directing acting antiviral treatment for hepatitis C, alcohol abuse, and the rising obesity epidemic in the country [6]. However, a complete analysis of all causes of liver disease and HCC among adult liver transplant recipients in the U.S. has not been reported. In 2020, the Organ Procurement and Transplant Network (OPTN) reported that the major indication for liver transplants in 2018 was for other/unknown causes [7], accounting for 33.9% of liver transplants. Although unsubstantiated, this has led to speculation that many in the other/unknown category represent patients with NAFLD [8]. Liver transplants for HCC were also frequently reported as a separate category and not under the liver disease causing HCC. This leads to an undercount of the actual burden of the underlying etiology of liver disease treated by liver transplantation. The purpose of this study is to conduct an in-depth analysis of all causes of liver disease and their associated complications including HCC among liver transplants recipients from 2010 to 2019.

## Materials and methods

The data for this analysis was obtained from the Organ Procurement and Transplantation Network (OPTN) STAR database, a national database operated by the United Network of Organ Sharing (UNOS) under contract with the U.S. Department of Health and Human Services that contains all the data in the U.S. on organ transplantation including waiting list, organ donation, and transplantation. All the data contained in the OPTN database are fully anonymized and the authors had no access to any identifying information. The study cohort included all the adult patients aged 18 years and older (66,719) who received liver transplants between January 1, 2010 and December 31, 2019. The causes of liver disease (S1 Table) were categorized by combining the diagnoses entered into the database as primary and secondary diagnoses at initial waitlist registration, final diagnosis entered at time of transplant, and positive serology for hepatitis B (hepatitis B surface antigen, HBsAg) and hepatitis C (hepatitis C antibody). When there was more than one diagnosis, the cause of liver disease was categorized by prioritizing the final diagnosis at time of transplant, then primary diagnosis at time of waitlist registration, then secondary diagnosis at time of waitlist registration. The cause of liver disease was categorized under HCV and hepatitis B if the patients were hepatitis C antibody positive and hepatitis B surface antigen positive, respectively. The etiology of HCC was based on the liver diseases listed as primary or secondary diagnoses at waitlist, final diagnosis at time of transplant, and serology for hepatitis B and C. HCC and cryptogenic cirrhosis were only listed as separate categories when there was no known associated liver disease. Recipients who had no diagnosis information in the database were grouped under a category named 'unspecified cause'.

Recipients were analyzed by gender to assess trends in liver transplants and whether there were different disease burdens resulting in liver transplantation between men and women, and BMI at time of waitlist registration to assess the potential effect of the obesity epidemic.

## Statistical analysis

Statistical analysis was performed using SAS v9.4 software. Differences between the proportions of liver transplant performed each year for the various causes of liver disease compared with 2010 were calculated using a Z-test for two proportions. P-values and related confidence intervals (CIs) were reported accordingly. Linear regression analysis was conducted in SAS for trend over ten years, and the p-values and parameter estimates were reported accordingly.

## Results

### Trends in etiology of liver disease and HCC among adult liver transplant recipients from 2010 to 2019

The number of adult liver transplants per year has steadily increased from 5,731 in 2010 to 8,345 in 2019 (45.6% increase). Between 2010 and 2019, proportion of liver transplant recipients with BMI $\geq$ 30 has increased from 37.0% to 40.8% [3.8%, (95% CI 2.2% to 5.4%), p<0.001], and proportion of recipients with BMI $\geq$ 35 has increased from 13.8% to 16.7% [2.9%, (95% CI 1.7% to 4.1%), p<0.001] (Table 1).

Among adults who received liver transplants in 2019, the number and proportion of liver transplants by etiology of liver disease and HCC in descending order of frequency were alcohol-associated liver disease (2,399 or 28.8%), NAFLD (1,795 or 21.5%), HCV (1,563 or 18.7%), cholestatic liver disease (641 or 7.7%), hepatitis B with or without hepatitis C or hepatitis D coinfection (355 or 4.3%), cryptogenic cirrhosis (259 or 3.1%), autoimmune liver disease (234 or 2.8%), metabolic liver disease (243 or 2.9%), acute hepatic necrosis unrelated to hepatitis B or hepatitis C (169 or 2.1%), unspecified causes of HCC (132 or 1.6%), graft failure (142 or 1.7%), benign hepatic tumors (103 or 1.2%), non-HCC malignant liver tumors (93 or 1.1%), Budd-Chiari syndrome (30 or 0.4%), miscellaneous causes (32 or 0.4%), and unspecified causes (145 or 1.7%) (Table 1).

The proportion of liver transplants for hepatitis C has declined significantly. Although hepatitis C associated liver disease and HCC was the leading cause of liver transplants from 2010 to 2017, it dropped to number two behind alcohol associated liver disease in 2018 and to number three behind NAFLD among the most common causes of liver transplants in 2019. Between 2010 and 2019, the proportion of liver transplants for hepatitis C associated liver disease and HCC declined by more than half from 44.5% to 18.7% [25.8%, (95% CI 24.3% to 27.3%), p<0.001].

During the same period, the proportions of liver transplants for liver disease and HCC related to ALD and NAFLD have doubled from 12.7% to 28.8% [16.1%, (95% CI 14.8% to 17.4%), p<0.001], and from 9.1% to 21.5% [12.4%, (95% CI 11.2% to 13.5%), p<0.001], respectively. Proportion of liver transplants for graft failure and benign hepatic tumors have also increased from 0.2% to 1.7% [1.5%, (95% CI 1.2% to 1.8%), p<0.001] and from 0.7% to 1.2% [0.5%, (95% CI 0.2% to 0.8%), p = 0.003] respectively (Table 1).

Between 2010 and 2019, the proportions of liver transplants for hepatitis B associated liver disease and HCC decreased from 5.7% to 4.3% [1.4%, (95% CI 0.7% to 2.2%), p<0.001], cryptogenic cirrhosis from 4.7% to 3.1% [1.6%, (95% CI 0.9% to 2.3%), p<0.001], acute hepatic necrosis unrelated to hepatitis B or C from 3.1% to 2.1% [1.0%, (95% CI 0.5% to 1.6%), p<0.001], and unspecified causes of liver disease or cirrhosis from 3% to 1.7% [1.3%, (95% CI

**Table 1. Causes of liver disease and HCC among adult liver transplant recipients in the U.S. from 2010 to 2019 by frequency and percentage.**

| | 2010 | 2011 | 2012 | 2013 | 2014 | 2015 | 2016 | 2017 | 2018 | 2019 |
|---|---|---|---|---|---|---|---|---|---|---|
| **Total # of liver transplants (%)** | 5731 | 5806 | 5731 | 5921 | 6200 | 6547 | 7268 | 7483 | 7687 | 8345 |
| **BMI ≥30 at listing for LT** | 2120 (37.0%) | 2130 (36.7%) | 2189 (38.2%) | 2181 (36.8%) | 2374 (38.3%) | 2425 (37.0%) | 2838 (39.1%)† | 3056 (40.8%)† | 3038 (39.5%)† | 3406 (40.8%)† |
| **BMI ≥35 at listing for LT** | 792 (13.8%) | 819 (14.1%) | 814 (14.2%) | 860 (14.5%) | 893 (14.4%) | 972 (14.9%) | 1165 (16.0%)† | 1276 (17.1%)† | 1257 (16.4%)† | 1391 (16.7%)† |
| **Causes of liver disease** | | | | | | | | | | |
| **HCV** ** | 2548 (44.5%) | 2518 (43.4%) | 2613 (45.6%) | 2528 (42.7%) | 2623 (42.3%)† | 2468 (37.7%)† | 2247 (30.9%)† | 2124 (28.4%)† | 1857 (24.2%)† | 1563 (18.7%)† |
| **ALD** * | 727 (12.7%) | 764 (13.2%) | 754 (13.2%) | 811 (13.7%) | 917 (14.8%)† | 1158 (17.7%)† | 1496 (20.6%)† | 1694 (22.6%)† | 1920 (24.5%)† | 2399 (28.8%)† |
| **NAFLD** * | 519 (9.1%) | 534 (9.2%) | 595 (10.4%)† | 698 (11.8%)† | 792 (12.8%)† | 912 (13.9%)† | 1264 (17.4%)† | 1370 (18.3%)† | 1544 (20.1%)† | 1795 (21.5%)† |
| **Cholestatic** * | 481 (8.4%) | 544 (9.4%) | 464 (8.1%) | 492 (8.3%) | 480 (7.7%) | 551 (8.4%) | 641 (8.8%) | 651 (8.7%) | 591 (7.7%) | 641 (7.7%) |
| **HBV**\*\*\* | 325 (5.7%) | 354 (6.1%) | 296 (5.2%) | 323 (5.5%) | 324 (5.2%) | 298 (4.6%)† | 315 (4.3%)† | 327 (4.4%)† | 346 (4.5%)† | 355 (4.3%)† |
| **Cryptogenic** * | 270 (4.7%) | 270 (4.7%) | 208 (3.6%)† | 208 (3.5%)† | 180 (2.9%)† | 210 (3.2%)† | 264 (3.6%)† | 245 (3.3%)† | 270 (3.5%)† | 259 (3.1%)† |
| **Autoimmune** * | 148 (2.6%) | 155 (2.7%) | 171 (3.0%) | 167 (2.8%) | 147 (2.4%) | 177 (2.7%) | 212 (2.9%) | 203 (2.7%) | 215 (2.8%) | 234 (2.8%) |
| **Metabolic** * | 148 (2.6%) | 170 (2.9%) | 167 (2.9%) | 164 (2.8%) | 181 (2.9%) | 171 (2.6%) | 218 (3.0%) | 224 (3.0%) | 200 (2.6%) | 243 (2.9%) |
| **Acute hepatic necrosis (non-HBV/HCV)** | 180 (3.1%) | 149 (2.6%) | 137 (2.4%)† | 146 (2.5%)† | 155 (2.5%)† | 143 (2.2%)† | 153 (2.1%)† | 150 (2.0%)† | 165 (2.2%)† | 179 (2.1%)† |
| **Unspecified causes of HCC** | 77 (1.3%) | 62 (1.1%) | 68 (1.2%) | 91 (1.5%) | 100 (1.6%) | 80 (1.2%) | 63 (0.9%)† | 99 (1.3%) | 108 (1.4%) | 132 (1.6%) |
| **Graft failure** | 10 (0.2%) | 14 (0.2%) | 41 (0.7%)† | 65 (1.1%)† | 88 (1.4%)† | 100 (1.5%)† | 125 (1.7%)† | 104 (1.4%)† | 148 (1.9%)† | 142 (1.7%)† |
| **Benign hepatic tumors** | 37 (0.7%) | 31 (0.5%) | 28 (0.5%) | 29 (0.5%) | 33 (0.5%) | 57 (0.9%) | 50 (0.7%) | 67 (0.9%) | 57 (0.7%( | 103 (1.2%)† |
| **Other malignant liver tumors** | 46 (0.8%) | 52 (0.9%) | 46 (0.8%) | 58 (1.0%) | 59 (1.0%) | 73 (1.1%) | 45 (0.6%) | 69 (0.9%) | 73 (1.0%) | 93 (1.1%) |
| **Budd-Chiari** | 27 (0.5%) | 27 (0.5%) | 14 (0.2%)† | 22 (0.4%) | 13 (0.2%)† | 22 (0.3%) | 22 (0.3%) | 26 (0.4%) | 26 (0.3%) | 30 (0.4%) |
| **Miscellaneous causes** | 16 (0.3%) | 21 (0.4%) | 25 (0.4%) | 29 (0.5%) | 26 (0.4%) | 30 (0.5%) | 38 (0.5%)† | 25 (0.3%) | 33 (0.4%) | 32 (0.4%) |
| **Unspecified causes of liver disease/cirrhosis** | 172 (3.0%) | 141 (2.4%) | 104 (1.8%)† | 90 (1.5%)† | 82 (1.3%)† | 97 (1.5%)† | 115 (1.6%)† | 105 (1.4%)† | 134 (1.7%)† | 145 (1.7%)† |

*± HCC.

**± HCC/ALD.

\*\*\*± HCC/ALD/HCV/HDV (S2 Table).

† significantly different from 2010 (p<0.05).

Miscellaneous (trauma, congenital hepatic fibrosis, hyperalimentation induced, drug/industrial exposure related cirrhosis).

0.8% to 1.8%), p<0.001]. There were no changes in the proportion of liver transplants for liver disease and HCC associated with cholestatic liver disease, autoimmune liver disease, metabolic liver disease, unspecified causes of HCC, other malignant liver tumors, Budd-Chiari syndrome, and miscellaneous causes from 2010 to 2019 (Table 1).

In a linear regression analysis, the major changes in the trend in the causes of liver disease and HCC among adult transplant recipients between 2010 and 2019 were proportions of liver transplants for HCV (-2.98 [95% CI -3.85 to -2.11]), ALD (1.81 [95% CI 1.35 to 2.27]), and NAFLD (1.49 [95% CI 1.28 to 1.70]) (Table 2). ALD (2.06 [95% CI 1.54 to 2.58]) showed a greater increase in men relative to NAFLD (1.29 [95% CI 1.06 to 1.51]), while NAFLD (1.83

**Table 2. Linear regression slope estimates by year and p-values for proportion of liver transplants caused by each disease etiology in general population, men, and women.**

| | General | | Male | | Female | |
|---|---|---|---|---|---|---|
| | Linear Regression Parameter (95% CI) | P-value | Linear Regression Parameter (95% CI) | P-value | Linear Regression Parameter (95% CI) | P-value |
| **HCV** | -2.98 | p<0.001† | -3.17 | p<0.001† | -2.53 | p<0.001† |
| | (-3.85 to -2.11) | | (-4.14 to -2.20) | | (-3.17 to -1.89) | |
| **ALD** | 1.81 | p<0.001† | 2.06 | p<0.001† | 1.4 | p<0.001† |
| | (1.35 to 2.27) | | (1.54 to 2.58) | | (0.97 to 1.83) | |
| **NAFLD** | 1.49 | p<0.001† | 1.29 | p<0.001† | 1.83 | p<0.001† |
| | (1.28 to 1.70) | | (1.06 to 1.51) | | (1.59 to 2.06) | |
| **Cholestatic** | -0.08 | p = 0.21 | -0.08 | p = 0.22 | -0.13 | p = 0.18 |
| | (-0.21 to 0.05) | | (-0.21 to 0.06) | | (-0.34 to 0.07) | |
| **HBV** | -0.19 | p<0.001† | -0.21 | p<0.001† | -0.15 | p = 0.05† |
| | (-0.27 to -0.12) | | (-0.30 to -0.12) | | (-0.30 to -0.002) | |
| **Cryptogenic** | 0.14 | p = 0.02† | -0.11 | p = 0.08 | -0.2 | p = 0.005† |
| | (-0.26 to -0.03) | | (-0.25 to 0.02) | | (-0.32 to -0.08) | |
| **Autoimmune** | 0.01 | p = 0.5 | 0.01 | p = 0.55 | -0.03 | p = 0.56 |
| | (-0.03 to 0.05) | | (-0.04 to 0.06) | | (-0.13 to 0.07) | |
| **Metabolic** | 0.01 | p = 0.6 | 0.02 | p = 0.30 | -0.02 | p = 0.72 |
| | (-0.04 to 0.05) | | (-0.03 to 0.08) | | (-0.14 to 0.10) | |
| **Acute Hepatic Necrosis (Non-HBV** | -0.1 | p = 0.002† | 0.01 | p = 0.49 | -0.32 | p<0.001† |
| | (-0.14 to -0.04) | | (-0.03 to 0.06) | | (-0.42 to -0.23) | |
| **Other/Unspecified Causes of HCC** | 0.02 | p = 0.56 | 0.04 | p = 0.15 | -0.03 | p = 0.49 |
| | (-0.05 to 0.08) | | (-0.02 to 0.11) | | (-0.13 to 0.07) | |
| **Graft Failure** | 0.19 | p<0.001† | 0.16 | p<0.001† | 0.23 | p<0.001† |
| | (0.12 to 0.26) | | (0.09 to 0.23) | | (0.14 to 0.33) | |
| **Benign Hepatic Tumors** | 0.06 | p = 0.01† | 0.03 | p = 0.06 | 0.1 | p = 0.02† |
| | (0.02 to 0.10) | | (-0.001 to 0.06) | | (0.02 to 0.19) | |
| **Other Malignant Liver Tumors** | 0.02 | p = 0.32 | 0.05 | p = 0.05 | -0.04 | p = 0.14 |
| | (-0.02 to 0.05) | | (-0.001 to 0.1) | | (-0.10 to 0.02) | |
| **Budd-Chiari** | -0.009 | p = 0.38 | -0.01 | p = 0.41 | -0.01 | p = 0.52 |
| | (-0.03 to 0.01) | | (-0.03 to 0.01) | | (-0.05 to 0.03) | |
| **Miscellaneous** | 0.006 | p = 0.5 | 0.01 | p = 0.22 | -0.02 | p = 0.50 |
| | (-0.01 to 0.03) | | (-0.009 to 0.03) | | (-0.03 to 0.01) | |
| **Other/ Unspecified** | -0.11 | p = 0.05 | -0.12 | p = 0.05† | -0.09 | p = 0.05 |
| | (-0.2 to 0.001) | | (-0.24 to -0.003) | | (-0.18 to 0.001) | |

†Significant trend.

[95% CI 1.59 to 2.06]) showed a greater increase in women than ALD (1.4 [95% CI 0.97 to 1.83]) (Table 2).

## Trends in etiology of liver disease among men who received liver transplants from 2010 to 2019

The ratio of men and women who received liver transplant each year was stable at about 1.8:1. Male liver transplant recipients increased from 3,769 in 2010 to 5,314 in 2019 (41% increase). Between 2010 and 2019, there was a significant increase in men with BMI ≥ 30 (from 36.0% to 41.5%) [5.5%, (95% CI 3.5% to 7.5%), p<0.001] and BMI ≥ 35 (from 12.4% to 15.7%) [3.3%,

(95% CI 1.9% to 4.7%), p<0.001]. Among men who received liver transplants, the five leading causes in 2019 in decreasing order of frequency were liver disease and HCC associated with alcohol associated liver disease, hepatitis C, NAFLD, cholestatic liver disease and hepatitis B (Table 3).

Since 2019, alcohol associated liver disease and HCC have surpassed hepatitis C as the leading cause of liver disease among men who received liver transplants. Between 2010 and 2019, the proportion of liver transplants for liver disease and HCC associated with ALD increased from 14.5% to 33.1 [18.6%, (95% CI 16.9% to 20.3%), p<0.001]. Hepatitis C-related liver disease and HCC remained the second most common cause of liver transplant in men in 2019, but it has dropped from 49.4% of liver transplants in 2010 to 21.9% in 2019 [27.5%, (95% CI 25.5% to 29.4%), p<0.001]. The proportion of liver transplants in men for liver disease and HCC associated with NAFLD has also doubled from 7.6% to 18.5% [10.9%, (95% CI 9.5% to 12.2%), p<0.001], between 2010 and 2019. Liver transplants for hepatitis B-related liver disease

**Table 3. Causes of liver disease and HCC among men who received liver transplant in the U.S. from 2010 to 2019 by frequency and percentage.**

| | 2010 | 2011 | 2012 | 2013 | 2014 | 2015 | 2016 | 2017 | 2018 | 2019 |
|---|---|---|---|---|---|---|---|---|---|---|
| **Total # of liver transplants (%)** | 3769 | 3871 | 3871 | 3901 | 4136 | 4346 | 4757 | 4864 | 4982 | 5314 |
| **BMI ≥30 at listing for LT** | 1358 (36.0%) | 1412 (36.5%) | 1463 (37.8%) | 1408 (36.1%) | 1573 (38.0%) | 1588 (36.5%) | 1874 (39.4%)† | 2017 (41.5%)† | 2020 (40.6%)† | 2207 (41.5%)† |
| **BMI ≥35 at listing for LT** | 468 (12.4%) | 507 (13.1%) | 493 (12.7%) | 510 (13.1%) | 534 (12.9%) | 582 (13.4%) | 711 (15.0%)† | 798 (16.4%)† | 796 (16.0%)† | 834 (15.7%)† |
| **Causes of liver disease** | | | | | | | | | | |
| **HCV** ** | 1861 (49.4%) | 1890 (48.8%) | 1963 (50.7%) | 1863 (47.8%) | 1985 (48.0%) | 1852 (42.6%)† | 1679 (35.3%)† | 1603 (33.0%)† | 1409 (28.3%)† | 1165 (21.9%)† |
| **ALD** * | 546 (14.5%) | 586 (15.1%) | 590 (15.2%) | 633 (16.2%)† | 705 (17.1%)† | 868 (20.0%)† | 1148 (24.1%)† | 1270 (26.1%)† | 1404 (28.2%)† | 1759 (33.1%)† |
| **NAFLD** * | 286 (7.6%) | 295 (7.6%) | 362 (9.4%)† | 379 (9.7%)† | 453 (11.0%)† | 487 (11.2%)† | 714 (15.0%)† | 757 (15.6%)† | 857 (17.2%)† | 982 (18.5%)† |
| **Cholestatic** * | 237 (6.3%) | 278 (7.2%) | 233 (6.0%) | 232 (6.0%) | 217 (5.3%)† | 288 (6.6%) | 306 (6.4%) | 298 (6.1%) | 279 (5.6%) | 305 (5.7%) |
| **HBV*** | 251 (6.7%) | 260 (6.7%) | 219 (5.7%) | 254 (6.5%) | 262 (6.3%) | 239 (5.5%)† | 251 (5.3%)† | 250 (5.1%)† | 254 (5.1%)† | 259 (4.9%)† |
| **Cryptogenic** * | 162 (4.3%) | 174 (4.5%) | 122 (3.2%)† | 118 (3.0%)† | 106 (2.6%)† | 140 (3.2%)† | 156 (3.3%)† | 153 (3.2%)† | 167 (3.4%)† | 155 (2.9%)† |
| **Autoimmune** * | 46 (1.2%) | 42 (1.1%) | 49 (1.3%) | 41 (1.1%) | 28 (0.7%)† | 54 (1.2%) | 65 (1.4%) | 54 (1.1%) | 65 (1.3%) | 65 (1.2%) |
| **Metabolic** * | 107 (2.8%) | 107 (2.8%) | 108 (2.8%) | 121 (3.1%) | 119 (2.9%) | 124 (2.9%) | 143 (3.0%) | 165 (3.4%) | 135 (2.7%) | 161 (3.0%) |
| **Acute hepatic necrosis (non-HBV/HCV)** | 46 (1.2%) | 46 (1.2%) | 34 (0.9%) | 44 (1.1%) | 46 (1.1%) | 49 (1.1%) | 53 (1.1%) | 43 (0.9%) | 58 (1.2%) | 81 (1.5%) |
| **Unspecified causes of HCC** | 55 (1.5%) | 48 (1.2%) | 43 (1.1%) | 53 (1.4%) | 66 (1.6%) | 54 (1.2%) | 46 (1.0%)† | 74 (1.5%) | 83 (1.7%) | 100 (1.9%) |
| **Graft failure** | 6 (0.2%) | 7 (0.2%) | 24 (0.6%)† | 41 (1.1%)† | 49 (1.2%)† | 53 (1.2%)† | 70 (1.5%)† | 49 (1.0%)† | 92 (1.9%)† | 78 (1.5%)† |
| **Benign hepatic tumors** | 13 (0.3%) | 8 (0.2%) | 8 (0.2%) | 7 (0.2%) | 9 (0.2%) | 10 (0.2%) | 10 (0.2%) | 22 (0.45%) | 17 (0.3%) | 34 (0.6%) |
| **Other malignant liver tumors** | 24 (0.6%) | 26 (0.7%) | 33 (0.9%) | 42 (1.1%)† | 32 (0.8%) | 48 (1.1%)† | 26 (0.55%) | 48 (1.0%) | 56 (1.1%)† | 67 (1.3%)† |
| **Budd-Chiari** | 15 (0.4%) | 10 (0.3%) | 6 (0.15%)† | 11 (0.3%) | 4 (0.1%)† | 9 (0.2%) | 6 (0.1%)† | 12 (0.25%) | 16 (0.3%) | 9 (0.2%)† |
| **Miscellaneous** | 6 (0.2%) | 9 (0.2%) | 15 (0.4%) | 16 (0.4%)† | 13 (0.3%) | 17 (0.4%)† | 20 (0.4%)† | 13 (0.3%) | 19 (0.4%) | 17 (0.3%) |
| **Unspecified causes of liver disease/cirrhosis** | 108 (2.9%) | 85 (2.2%) | 62 (1.6%)† | 46 (1.2%)† | 42 (1.0%)† | 54 (1.2%)† | 64 (1.3%)† | 53 (1.1%)† | 71 (1.4%)† | 77 (1.5%)† |

* ± HCC.

** ± HCC/ALD.

*** ± HCC/HCV/ALD/HDV.

† significantly different from 2010.

and HCC in men decreased from 6.7% to 4.9% [1.8%, (95% CI 0.8% to 2.8%), p<0.001] between 2010 and 2019 (Table 3).

## Trends in etiology of liver disease among women who received liver transplants from 2010 to 2019

The number of women who received liver transplants increased from 1,962 in 2010 to 3,031 in 2019 (54.5% increase). In 2010, more women than men had BMI ≥ 30 (38.8% vs 36.0%) [2.8%, (95% CI 0.2% to 5.5%), p = 0.04] and BMI ≥ 35 (16.5% vs 12.4%) [4.1%, (95% CI 2.2% to 6.1%), p<0.001]. By 2019, the percentage of women and men with BMI ≥ 30 was the same (41.5% vs 39.6%), but more women than men had BMI ≥ 35 (18.4% vs 15.7%) [2.7%, (95% CI 1.0% to 4.4%), p = 0.002]. (Table 4)

Similar to the trend seen in men, there was a significant decrease in proportion of liver transplants in women for hepatitis C-associated liver disease and HCC from 35% in 2010 to 13.1% in 2019 [21.9%, (95% CI 19.5% to 24.3%), p<0.001], whereas the proportions of liver transplants for liver disease and HCC associated with NAFLD and ALD more than doubled from 11.9% to 26.8% [14.9%, (95% CI 13.3% to 16.5%), p<0.001] and from 9.2% to 21.1% [11.9%, (95% CI 10.5% to 13.3%), p<0.001], respectively (Table 4). In 2017 NAFLD surpassed hepatitis C as the leading cause of liver disease in women receiving liver transplants, and in 2019 alcohol-associated liver disease surpassed HCV to become the second major cause of liver transplants in women. Among women who received liver transplants in 2019, the five leading causes of liver disease in decreasing order of frequency were liver disease and HCC associated with NAFLD, alcohol associated liver disease, hepatitis C, cholestatic liver disease, and autoimmune liver disease (Table 4).

## Etiology of HCC among adult liver transplant recipients between 2010 and 2019

From January 1, 2010 to December 31, 2019, 19,872 (31%) adult liver transplant recipients were diagnosed with HCC (S3 Table). From 2010 to 2019, the proportion of transplant recipients with HCC decreased from 33% to 29% in men [4.3%, 95% CI (2.4% to 6.2%), p<0.001] and from 20% to 15.5% in women [4.5%, 95% CI (2.9% to 6.1%), p<0.001]. Hepatitis C remained the leading cause of HCC, although it has declined from 67.8% in 2010 to 43.8% [24.0%, (95% CI 20.8% to 27.1%), p<0.001] in 2019. Liver transplants for HCC related to hepatitis B have seen no significant change. During the same period, liver transplants for HCC related to NAFLD and alcohol associated liver disease have both increased from 5.0% to 18.8% [13.8%, (95% CI 11.8% to 15.8%), p<0.001], and from 7.0% to 15.8% [8.8%, (95% CI 6.8% to 10.8%), p<0.001], respectively. Between 2010 and 2019, there were no significant changes in the proportion of liver transplants for HCC related to cholestatic liver disease (1.9%), cryptogenic cirrhosis (2.0%), autoimmune cirrhosis (1.0%), metabolic disease (1.0%), and Budd-Chiari (0.2%) (Table 5).

When analyzed by gender, hepatitis C remained the leading cause of HCC in both men and women who received liver transplants although it has dropped from 68.3% to 46.0% [22.3%, (95% CI 18.7% to 25.9%), p<0.001] in men and from 66.4% to 36.7% [29.7%, (95% CI 23.2% to 35.9%), p<0.001] in women between 2010 and 2019. In 2019, the second most common cause of HCC was ALD in men (19%), but in women it was NAFLD (30%). In women, ALD was not a common cause of HCC accounting for only 7.0% of liver transplants for HCC in 2019 (Table 5).

**Table 4. Causes of liver disease and HCC among women who received liver transplant in the U.S. from 2010 to 2019 by frequency and percentage.**

| | 2010 | 2011 | 2012 | 2013 | 2014 | 2015 | 2016 | 2017 | 2018 | 2019 |
|---|---|---|---|---|---|---|---|---|---|---|
| **Total # of liver transplants (%)** | 1962 | 1935 | 1860 | 2020 | 2064 | 2201 | 2511 | 2619 | 2705 | 3031 |
| **BMI ≥30 at listing for LT** | 762 (38.8%) | 718 (37.1%) | 726 (39.0%) | 773 (38.3%) | 801 (38.8%) | 837 (38.0%) | 964 (38.4%) | 1039 (39.7%) | 1018 (37.6%) | 1199 (39.6%) |
| **BMI ≥35 at listing for LT** | 324 (16.5%) | 312 (16.1%) | 321 (17.3%) | 350 (17.3%) | 359 (17.4%) | 390 (17.7%) | 454 (18.1%) | 478 (18.2%) | 461 (17.0%) | 557 (18.4%) |
| **Causes of liver disease** | | | | | | | | | | |
| **HCV** [**] | 687 (35.0%) | 628 (32.5%) | 650 (35.0%) | 665 (32.9%) | 638 (30.9%†) | 616 (28.0%)† | 568 (22.6%)† | 521 (19.9%)† | 448 (16.6%)† | 398 (13.1%)† |
| **ALD** [*] | 181 (9.2%) | 178 (9.2%) | 164 (8.8%) | 178 (8.8%) | 212 (10.3%) | 290 (13.2%)† | 348 (13.9%)† | 424 (16.2%)† | 516 (19.1%)† | 640 (21.1%)† |
| **NAFLD** [*] | 233 (11.9%) | 239 (12.4%) | 233 (12.5%) | 319 (15.8%)† | 339 (16.4%)† | 425 (19.3%)† | 550 (21.9%)† | 613 (23.4%)† | 687 (25.4%)† | 813 (26.8%)† |
| **Cholestatic** [*] | 244 (12.4%) | 266 (13.75%) | 231 (12.4%) | 260 (12.9%) | 263 (12.7%) | 263 (11.95%) | 335 (13.3%) | 353 (13.5%) | 312 (11.5%) | 336 (11.1%) |
| **HBV** [***] | 74 (3.8%) | 94 (4.9%) | 77 (4.1%) | 69 (3.4%) | 62 (3.0%) | 59 (2.7%)† | 64 (2.6%)† | 77 (2.9%) | 92 (3.4%) | 96 (3.2%) |
| **Cryptogenic** [*] | 108 (5.5%) | 96 (5.0%) | 86 (4.6%) | 90 (4.5%) | 74 (3.6%)† | 70 (3.2%)† | 108 (4.3%) | 92 (3.5%)† | 103 (3.8%)† | 104 (3.4%)† |
| **Autoimmune** [*] | 102 (5.2%) | 113 (5.8%) | 122 (6.6%) | 126 (6.2%) | 119 (5.8%) | 123 (5.6%) | 147 (5.85%) | 149 (5.7%) | 150 (5.55%) | 169 (5.6%) |
| **Metabolic** [*] | 41 (2.1%) | 63 (3.3%)† | 59 (3.2%)† | 43 (2.1%) | 62 (3.0%) | 47 (2.1%) | 75 (3.0%) | 59 (2.25%) | 65 (2.4%) | 82 (2.7%) |
| **Acute hepatic necrosis (non-HBV/HCV)** | 134 (6.8%) | 103 (5.3%)† | 103 (5.5%) | 102 (5.1%)† | 109 (5.3%)† | 94 (4.3%)† | 100 (4.0%)† | 107 (4.1%)† | 107 (4.0%)† | 98 (3.2%)† |
| **Unspecified causes of HCC** | 22 (1.1%) | 14 (0.7%) | 25 (1.3%) | 38 (1.9%)† | 34 (1.7%) | 26 (1.2%) | 17 (0.7%) | 25 (0.95%) | 25 (0.9%) | 32 (1.1%) |
| **Graft failure** | 4 (0.2%) | 7 (0.4%) | 17 (0.9%)† | 24 (1.2%)† | 39 (1.9%)† | 47 (2.1%)† | 40 (1.6%)† | 45 (1.7%)† | 40 (1.5%)† | 69 (2.3%)† |
| **Benign hepatic tumors** | 24 (1.2%) | 23 (1.2%) | 20 (1.1%) | 22 (1.1%) | 24 (1.2%) | 47 (2.1%)† | 40 (1.6%) | 45 (1.7%) | 40 (1.5%) | 69 (2.3%)† |
| **Other malignant liver tumors** | 22 (1.1%) | 26 (1.3%) | 13 (0.7%) | 16 (0.8%) | 27 (1.3%) | 25 (1.1%) | 19 (0.8%) | 21 (0.8%) | 17 (0.6%) | 26 (0.9%) |
| **Budd-Chiari** | 12 (0.6%) | 17 (0.9%) | 8 (0.4%) | 11 (0.5%) | 9 (0.4%) | 13 (0.6%) | 16 (0.6%) | 14 (0.5%) | 10 (0.4%) | 21 (0.7%) |
| **Miscellaneous** | 10 (0.5%) | 12 (0.6%) | 10 (0.5%) | 13 (0.6%) | 13 (0.6%) | 13 (0.6%) | 18 (0.7%) | 12 (0.5%) | 14 (0.5%) | 15 (0.5%) |
| **Unspecified causes of liver disease/cirrhosis** | 64 (3.3%) | 56 (2.9%) | 42 (2.3%) | 44 (2.2%)† | 40 (1.9%)† | 43 (1.9%)† | 51 (2.0%)† | 52 (2.0%)† | 63 (2.3%) | 68 (2.2%)† |

[*] ± HCC.

[**] ± HCC/ALD.

[***] ± HCC/ALD/HBV/HDV.

† significant different from 2010.

## Discussion

The major indication for liver transplantation in 2018 was listed as other/unknown causes (33.9%) in the 2020 Organ Procurement and Transplant Network (OPTN) Report [7]. But in this study, when all causes of liver disease and HCC were examined by compiling all the diagnosis and HBV and HCV serologic data entries in the OPTN STAR database from time of initial waitlist registration to time of transplantation, we found only a very small proportion (1.7%) of liver transplant in adults had unknown causes of liver disease in 2018 and 2019.

Our study found that although hepatitis C related liver disease and HCC was the leading cause for liver transplants in the U.S. and accounted for half of liver transplants in men and a third of the liver transplants in women between 2010 to 2014, there has been a dramatic and sustained decline since then. Since 2010, the proportion of liver transplants for hepatitis C

**Table 5. Liver disease associated with HCC among men and women who received LT in the U.S. between 2010 and 2019.**

| | Total | | Male | | Female | |
|---|---|---|---|---|---|---|
| | **2010** | **2019** | **2010** | **2019** | **2010** | **2019** |
| **# and % of LT with HCC** | 1634 (28.5%) | 1992 (23.8) † | 1241(32.9%) | 1521 (28.6) † | 393 (20%) | 471 (15.5%)† |
| **Etiology of HCC** | | | | | | |
| **HCV[1]** | 1108 (67.8%) | 873 (43.8%)† | 847(68.3%) | 700 (46.0%)† | 261 (66.4%) | 173 (36.7%)† |
| **ALD** | 115 (7.0%) | 315 (15.8%)† | 102 (8.2%) | 282 (18.5%)† | 13 (3.3%) | 33 (7.0%)† |
| **NAFLD** | 82 (5.0%) | 374 (18.8%)† | 55 (4.4%) | 234 (15.4%)† | 27 (6.9%) | 140 (29.7%)† |
| **Cholestatic** | 31 (1.9%) | 38 (1.9%) | 17 (1.4%) | 18 (1.2%) | 14 (3.6%) | 20 (4.25%) |
| **HBV[2]** | 139 (8.5%) | 173 (8.7%) | 110 (8.9%) | 138 (9.1%) | 29 (7.4%) | 35 (7.4%) |
| **Cryptogenic** | 49 (3.0%) | 40 (2.0%) | 33 (2.7%) | 28 (1.8%) | 16 (4.1%) | 12 (2.5%) |
| **Autoimmune** | 14 (0.9%) | 21 (1.05%) | 5 (0.4%) | 5 (0.3%) | 9 (2.3%) | 16 (3.4%) |
| **Metabolic** | 17 (1.0%) | 21 (1.05%) | 15 (1.2%) | 15 (1.0%) | 2 (0.5%) | 6 (1.3%) |
| **Budd-Chiari** | 1 (0.1%) | 5 (0.25%) | 1 (0.1%) | 1 (0.1%) | 0 (0%) | 4 (0.85%) |
| **Miscellaneous** | 1 (0.1%) | 0 (0%) | 1 (0.1%) | 0 (0%) | 0 (0%) | 0 (0%) |
| **Unspecified Causes of HCC** | 77 (4.7%) | 132 (6.6%)† | 55 (4.4%) | 100 (6.6%)† | 22 (5.6%) | 32 (6.8%) |

[1] ± ALD.

[2] ± HCV/ALD/HDV.

† significantly different from 2010.

associated complications including HCC has decreased by 33.2% in 2017 and 55.7% in 2019 in men, and by 43.1% in 2017 and 62.6% in 2019 in women. The decline was even greater for hepatitis C without associated HCC (S4 Table). The declining numbers and proportions of liver transplants for liver disease and HCC associated with hepatitis C is in concordance with the reports of a similar decrease in proportion of patients with hepatitis C enrolled on the liver transplant waitlist [5]. This decline in liver transplant for hepatitis C complication is contemporaneously associated with increased screening and the introduction of the highly effective, curative direct acting antiviral therapies for hepatitis C [6]. There was also a decrease in the proportion of liver transplants for liver disease and HCC related to hepatitis B between 2010 and 2019. This may, likewise, be attributed to antiviral therapy and screening programs for chronic hepatitis B.

With the decline in liver transplants for HCV, the proportion of liver transplants for liver disease and HCC associated with NAFLD has more than doubled from 7.6% to 18.5% in men, and from 11.9% to 26.8% in women between 2010 and 2019. In 2017, NAFLD surpassed hepatitis C as the leading etiology of liver disease among women who received liver transplants and remained the third most common etiology of liver disease among men who received liver transplants. The increase in liver transplant for NAFLD is associated with a growing obesity crisis [6]. Between 2010 and 2019, proportion of male transplant recipients who were severely obese (BMI ≥ 35) increased from 12.4% to 15.7%, and from 16.5% to 18.4% in women. Past studies have suggested that many transplant recipients with cryptogenic cirrhosis may represent uncounted cases of NAFLD [9, 10] and should be included as a part of the NAFLD cohort. We found among the 259 (3.1% of total) transplants for cryptogenic cirrhosis in 2019, only 32 (12.4%) recipients had a BMI ≥ 35, and among the 145 (1.7% of total) transplants with unspecified liver disease, only 6 (4.1%) had a BMI ≥ 35. Therefore, they would be unlikely contribute to a significant increase in cases of NAFLD. Liver transplants for liver disease and HCC associated with NAFLD will likely continue to grow as a result of the high prevalence of severe and morbid obesity in the U.S. population. The CDC reported among adults living in the U.S.

between 2017–2018, the age adjusted prevalence of obesity in adults (BMI ≥ 30) was 42.4%, and 6.9% of men and 11.5% of women were morbidly obese (BMI ≥ 40) [11].

While the proportion of liver transplants for hepatitis C has been declining, the proportion of liver transplants for liver disease and HCC related to ALD has increased by 130% in both men (from 14.5% to 33.1%) and women (from 9.2% to 21.1%) between 2010 and 2019. ALD has become the leading indication for liver transplants in men and the second major indication for liver transplants in women in 2019. This trend is associated with the rising alcohol use in the U.S. population [12, 13]. Although men are traditionally heavier drinkers than women, alcohol consumption is rising in women as well. The National Epidemiologic Survey on Alcohol and Related Conditions reported increases in alcohol use especially among women in addition to older adults and racial/ethnic minorities [14, 15]. The increase in liver transplants for ALD has also been attributed to broader acceptance of alcoholic liver disease for liver transplants and the relaxation of the mandatory six-month abstinence period prior to transplantation by many transplant centers across the country [16]. Although our study included transplant recipients with HCC or cirrhosis (without HCC) under the same underlying etiology (Tables 1–3), a similar trend in liver transplants for HCV, ALD and NAFLD was found in both recipients without HCC and recipients with HCC (S4 Table).

HCC is a common indication for liver transplantation, although it has declined from 33% to 29% of the transplants in men, and from 20% to 15% among women who received liver transplants between 2010 and 2019. Hepatitis C was the most common cause of HCC among liver transplant recipients, but it has decreased from 68.3% to 46.0% in men and from 66.4% to 36.7% in women between 2010 and 2019. The proportion of liver transplants for HCC caused by ALD has increased by 130% in men (from 8.2% to 18.5%) between 2010 and 2019, although it remained a less common cause of HCC in women. Proportion of liver transplants for HCC caused by NAFLD showed the greatest increase from 4.4% to 15.4% in men and from 6.9% to 29.7% in women between 2010 and 2019. Our findings are consistent with a recent study that in the first quarter of 2019, NASH has surpassed HCV (including HCV+ALD) among women with HCC on the liver transplant waitlist, but HCV remained the major etiology among men with HCC on the liver transplant waitlist in 2019 [5]. Hepatitis B was the second most common cause of HCC in men and women in 2010, but has since dropped to fourth in men and third in women.

Between 2010 and 2019, there was a decrease in the proportions of liver transplants for acute hepatic necrosis (unrelated to hepatitis B or C), cryptogenic cirrhosis and in the category of unspecified diagnosis. Although the reason for the decrease in acute hepatic necrosis is unclear, the latter two decreasing trends likely reflect better diagnosis and coding entries into the UNOS database. There were also increases in liver transplants for graft failure and benign hepatic tumors that were not been previously reported. In the past 10 years, there were no significant changes in the proportions of liver transplants for cholestatic liver disease, autoimmune liver disease, metabolic liver disease, unspecified causes of HCC, other malignant tumors, Budd-Chiari, and in the several diseases we placed in the miscellaneous category.

From 2010 to 2019, about 75% of liver transplants in adults and 87% of liver transplants for HCC each year were performed for complications of largely preventable or treatable causes (hepatitis C and hepatitis B, alcohol associated liver disease, and NAFLD). A public health response to improve awareness, prevention, early diagnosis, and treatment is much needed to help to reduce and eliminate the burden of liver disease and HCC caused by these four diseases. Hepatitis C is an example of what can be achieved through a national public health campaign. Once the most common indication accounting for almost half of liver transplants each year, the continued decline in the number and proportion of men and women transplanted or on the transplant waitlist for hepatitis C demonstrates that with a national campaign to

increase hepatitis C screening and treatment, it is feasible to eliminate or reduce its complications and the need for transplantation. Continued declines in liver transplants for HCC related complications can occur by implementing the 2020 US Preventive Services Task Force (USPSTP) recommendations for universal screening, preventing new infections through harm reduction and needle exchange, and widening access to direct acting antiviral treatment by eliminating fibrosis restrictions [17–20]. Although a decrease in transplants for hepatitis B was seen, there was no change in transplants for HCC associated with hepatitis B. The risks of hepatitis B progression to HCC and cirrhosis can likewise be combated through hepatitis B immunization, public health campaigns to increase screening of at-risk populations including high prevalence foreign-born persons (according to USPSTF recommendations), and long-term disease monitoring and antiviral treatment [21–23]. The rising trend in liver transplants for NAFLD and alcohol-associated liver disease is concerning. Since there is no targeted pharmacological treatment for NAFLD [24], public health campaigns to promote healthy eating and exercise beginning from early childhood, and increased participation of healthcare providers in obesity screening and behavioral counselling are necessary to decrease risk of NAFLD complications [25–27]. Likewise, interventions to reduce alcohol consumption including awareness campaigns and regulation of advertising and pricing, alcohol misuse screenings, and clinical interventions [28] are needed to reduce the future burden of end stage liver disease and HCC caused by alcohol associated liver disease.

## Strength and limitations

While prior studies frequently reported HCC as a separate category and not under the liver disease that caused HCC or have focused on hepatitis C, NAFLD or ALD among patients on the waitlist for liver transplant, this is the only study that analyzed all causes of liver disease and HCC among adults who received liver transplants. The limitations of our study include that we restricted our analysis to transplant recipients, so our study does not reflect overall numbers and prevalence of these liver conditions among patients on the transplant waiting list. In addition, while we tried our best to categorize patients into disease categories, a small number of patients where multiple conditions were present, we included them only under their primary diagnosis. We are further limited by the accuracy of data entry and availability of information recorded in the database–for instance we lacked information on metabolic syndrome and risk factors of NAFLD. Despite these limitations, our study provides the first complete analysis of all causes of liver disease among liver transplant recipients between 2010 and 2019.

## Conclusions

When all causes of liver disease and HCC were examined, the proportion of liver transplants for unspecified indications in 2019 was only 1.7%. About 75% of liver transplants in adults and 87% of liver transplants for HCC each year were performed for complications of largely preventable or treatable diseases (hepatitis C and B, alcohol associated liver disease and NAFLD), underlying the importance of public health interventions in raising awareness, prevention, early diagnosis and treatment to prevent disease progression to end stage liver disease and HCC. The decline in liver transplantation for HCV demonstrates the feasibility to reduce the burden of end stage liver disease caused by hepatitis C through an increase in screening and treatment efforts.

## Supporting information

**S1 Table. Diagnosis categorization criteria (methodology).**
(DOCX)

**S2 Table. HBV transplants and co-infection with HCV or HDV, by year, frequency and percent.**
(DOCX)

**S3 Table. Underlying etiology of HCC among adult liver transplant recipients from 2010 to 2019, by frequency.**
(DOCX)

**S4 Table. Etiology of liver disease among liver transplant recipients with or without HCC by year, frequency and percentage.**
(DOCX)

## Acknowledgments

This work was based on data collected and provided by the Organ Procurement and Transplant Network which is supported in part by Health Resources and Services Administration contract 234-2005-370011C. The content is the responsibility of the authors alone and does not necessarily reflect the views or policies of the Department of Health and Human Services, nor does mention of trade names, commercial products, or organizations imply endorsement by the U.S. Government.

## Author Contributions

**Conceptualization:** Sonia Wang, Mehlika Toy, Thi T. Hang Pham, Samuel So.

**Data curation:** Sonia Wang, Mehlika Toy, Thi T. Hang Pham, Samuel So.

**Formal analysis:** Sonia Wang, Mehlika Toy, Thi T. Hang Pham, Samuel So.

**Methodology:** Sonia Wang, Mehlika Toy, Thi T. Hang Pham, Samuel So.

**Project administration:** Sonia Wang, Mehlika Toy, Samuel So.

**Resources:** Sonia Wang, Mehlika Toy, Thi T. Hang Pham, Samuel So.

**Software:** Sonia Wang, Mehlika Toy, Samuel So.

**Supervision:** Sonia Wang, Samuel So.

**Validation:** Sonia Wang, Mehlika Toy, Thi T. Hang Pham, Samuel So.

**Visualization:** Sonia Wang, Mehlika Toy, Thi T. Hang Pham, Samuel So.

**Writing – original draft:** Sonia Wang, Mehlika Toy, Thi T. Hang Pham, Samuel So.

**Writing – review & editing:** Sonia Wang, Mehlika Toy, Thi T. Hang Pham, Samuel So.

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
