## [Decision Letter · Decision Letter 0]

21 Aug 2020

PONE-D-20-21329

Causes and trends in liver disease and hepatocellular carcinoma among men and women who received liver transplants in the U.S., 2010-2019.

PLOS ONE

Dear Dr. Samuel K So,

Thank you for submitting your manuscript to PLOS ONE. After careful consideration, we feel that it has merit but does not fully meet PLOS ONE’s publication criteria as it currently stands. Therefore, we invite you to submit a revised version of the manuscript that addresses the points raised during the review process.

Please submit your revised manuscript within 60 days. If you will need more time than this to complete your revisions, please reply to this message or contact the journal office at plosone@plos.org. Please include the following items when submitting your revised manuscript:

We look forward to receiving your revised manuscript.

Kind regards,

Gianfranco D. Alpini

Academic Editor

PLOS ONE

Journal Requirements:

2. In your ethics statement in the Methods section and in the online submission form, please provide additional information about the data used in your retrospective study. Specifically, please clarify whether all data were fully anonymized before you accessed them or if authors had access to identifying information.

3.Thank you for stating the following in the Acknowledgments Section of your manuscript:

[This work was based on data collected and provided by the Organ Procurement and Transplant

Network which is supported in part by Health Resources and Services Administration contract

234-2005-370011C.]

 [The author(s) received no specific funding for this work]

Please also provide an amended statement that declares *all* the funding or sources of support (whether external or internal to your organization) received during this study, as detailed online in our guide for authors at http://journals.plos.org/plosone/s/submit-now.  Please also include the statement “There was no additional external funding received for this study.” in your updated Funding Statement.

Reviewers' comments:

Reviewer's Responses to Questions

**Comments to the Author**

1. Is the manuscript technically sound, and do the data support the conclusions?

Reviewer #1: Yes

Reviewer #2: Yes

2. Has the statistical analysis been performed appropriately and rigorously? 

Reviewer #1: Yes

Reviewer #2: Yes

3. Have the authors made all data underlying the findings in their manuscript fully available?

Reviewer #1: Yes

Reviewer #2: Yes

4. Is the manuscript presented in an intelligible fashion and written in standard English?

Reviewer #1: Yes

Reviewer #2: Yes

5. Review Comments to the Author

Reviewer #1: Wang S and co-authors in the current paper evaluated the different causes for liver transplant (LT), in US, during the last 10 years on the base of OPTN data. Their results reflect with accuracy the changes occurring in the etiologies of liver diseases, including HCC, in the last decade in their country. This analysis is of interest and reports several information that deserve attention. These indications seem also useful to design public health system future strategies.

I just have minor comments:

1) How many HBV transplanted patients were actually HBV-HDV coinfected? Is it possible to retrieve this information from the database? This should be interesting since delta infection is considered aggressive and it is still in search of an adequate therapy. Please comment.

2) When a patient had a concurrent or previous infection with HCV (recently cured with DAA) and alcohol or NASH as cofactor, how was the final diagnosis ruled out?? Please comment.

3) Morbid obesity (>40 BMI) has been considered in the last decades as a possible absolute contraindication to LT by several centers. Do the authors believe that the increase of LT in NASH may be dependent, almost in part, by change of this attitude in recent years? Are there any data on this issue on OPTN database? Please comment.

Reviewer #2: In this paper the authors used the data from the national Organ Procurement and Transplant Network (OPTN) concerning the indication for liver transplants to examine all causes and trends in liver disease and hepatocellular carcinoma (HCC) among adults who received liver transplants in the past 10 years.

The study found NAFLD in 2017 in women and ALD in 2019 in men have surpassed HCV as the leading causes of liver disease and HCC among adults receiving liver transplants.

This is a timely and interesting paper. However different major points need to be clarified and discussed.

Major comments:

1) In this paper the authors merged two distinct categories of patients for each etiology, the ones with cirrhosis and the one with HCC. Moreover, the authors analyzed the changing trends of the etiologies of liver diseases in USA. Natural history of the liver diseases analyzed are very different, in particular with respect the risk of HCC development and the oncogenic potential of each etiologic factors. In my opinion having merged cirrhotic with HCC patients in the analysis of the time trends of the etiology may not reflect the ongoing scenario. Indeed, HCC development characterize the patients with the longer course of the diseases. Perhaps analyzing separately, the liver cirrhosis with respect the HCC can give different results and can be very informative with respect the real and future scenario of the epidemiology of liver diseases and liver transplantation. In case this is not feasible, please discuss this point and compare if possible these results with papers where the HCC and liver cirrhosis have been maintained separated.

2) The recent but largely accepted proposal to establish positive criteria to diagnose the Metabolic Associated Fatty Liver Disease (MAFLD), with new and easy diagnostic criteria need to be discussed. In particular, it will be nice to underline how the NAFLD diagnosis have been established with respect the use of alcohol, and if possible to evaluate how many of the NAFLD diagnosed patients also respect the MAFLD diagnosis.

3) Since this topic is very debated, I would like to see and compare the results of the papers facing the same topic of the changing etiology of transplanted HCC.

4) The diagnosis of NAFLD, ALD, cryptogenic, and metabolic liver cirrhosis should be carefully and precisely defined.

5) It should be mentioned whether other risk factors of NAFDL/MAFLD are known, e.g., DM II, metabolic syndrome, etc...

6) Many patients with cryptogenic liver cirrhosis had in their clinical history previous diagnosis of metabolic risk factors, which may also disappear as the liver disease evolves. When in the anamnesis we retrieve these factors, even if not present more, we prefer to attribute the etiology to NAFLD rather than to unknown causes. Please discuss whether anamnestic history can be considered to evaluate the etiologies.

6. PLOS authors have the option to publish the peer review history of their article (what does this mean?). If published, this will include your full peer review and any attached files.

Reviewer #1: No

Reviewer #2: No

---

## [Author Response · Author response to Decision Letter 0]

3 Sep 2020

Response to reviewers

Reviewer #1: 

1) How many HBV transplanted patients were actually HBV-HDV coinfected? Is it possible to retrieve this information from the database? This should be interesting since delta infection is considered aggressive and it is still in search of an adequate therapy. Please comment.

We have added Table S2 in the supplementary supporting information with a sub-analysis of HBV transplants including HDV coinfection information. The percentage of HBV transplanted patients with HDV coinfection was low, ranging from 1.2% to 4.8% per year.

2) When a patient had a concurrent or previous infection with HCV (recently cured with DAA) and alcohol or NASH as cofactor, how was the final diagnosis ruled out?? Please comment.

The national OPTN STAR liver transplant database only collects HCV diagnoses and HCV serology entered by the transplant centers. Unfortunately, it does not have HCV RNA data before 2018 or information on whether the patient has been cured by DAA at the time of waitlist registration or at transplantation. To give as accurate a depiction as possible of HCV in liver transplantation as a result of past or current infection, we included any patient with a positive HCV antibody or HCV diagnoses in the category of liver transplants associated with HCV even if they have alcohol or NASH as a cofactor. 

3) Morbid obesity (>40 BMI) has been considered in the last decades as a possible absolute contraindication to LT by several centers. Do the authors believe that the increase of LT in NASH may be dependent, almost in part, by change of this attitude in recent years? Are there any data on this issue on OPTN database? Please comment.

With the increasing incidence of NAFLD associated end stage liver disease and HCC, patients with NAFLD are increasingly accepted as candidates for liver transplantation in the U.S. Recent studies of the patients on the liver transplant waitlist showed that NAFLD is becoming the fastest growing indication for liver transplant. Despite the increased risk of perioperative complications, analysis of OPTN database for the association of BMI with patient and graft survival found surprisingly that lean NAFLD liver transplant recipients actually had lower graft and patient survival than all obese (class 1, 2 and 3 obesity) cohorts. (Satapathy SK, Jiang Y, Agbim U, et al. Posttransplant Outcome of Lean Compared With Obese Nonalcoholic Steatohepatitis in the United States: The Obesity Paradox. Liver Transpl. 2020;26(1):68-79. doi:10.1002/lt.25672) 

Reviewer #2: 

1) In this paper the authors merged two distinct categories of patients for each etiology, the ones with cirrhosis and the one with HCC. Moreover, the authors analyzed the changing trends of the etiologies of liver diseases in USA. Natural history of the liver diseases analyzed are very different, in particular with respect the risk of HCC development and the oncogenic potential of each etiologic factors. In my opinion having merged cirrhotic with HCC patients in the analysis of the time trends of the etiology may not reflect the ongoing scenario. Indeed, HCC development characterize the patients with the longer course of the diseases. Perhaps analyzing separately, the liver cirrhosis with respect the HCC can give different results and can be very informative with respect the real and future scenario of the epidemiology of liver diseases and liver transplantation. In case this is not feasible, please discuss this point and compare if possible these results with papers where the HCC and liver cirrhosis have been maintained separated.

Thank you for the excellent comments. The reason we included HCC in our analysis was because most of the reports in the U.S. have listed HCC regardless of the etiology of liver disease as a separate indication for liver transplant. By only including cirrhosis and not HCC, we felt it resulted in an under appreciation of the impact of all the complications associated with the underlying liver disease that resulted in liver transplantation. 

We have added Table S4 in the supplementary supporting information that separated the two groups, i.e., liver transplants associated with the various causes of liver disease with HCC and without HCC. 

When analyzed separately, the trend between 2010 and 2019 were:

HCV liver transplants without HCC declined by 67% (from 25.1% to 8.3%), and by 46% for HCV liver transplants with HCC (from 19.3% to 10.4%). 

ALD liver transplants without HCC increased by 2.3-fold (from 10.7% to 25%), and by 1.9-fold for ALD liver transplants with HCC (from 2% to 3.8%). 

NAFLD liver transplants without HCC increased 2.2-fold (from 7.6% to 17%), and by 3.2- fold for NAFLD liver transplants with HCC (from 1.4% to 4.5%)

We also added the following sentence in the discussion “Although our study included transplant recipients with HCC or cirrhosis (without HCC) under the same underlying etiology (Tables 1, 2, 3), a similar trend in liver transplants for HCV, ALD and NAFLD was found in both recipients without HCC and recipients with HCC (S4 Table).”

2) The recent but largely accepted proposal to establish positive criteria to diagnose the Metabolic Associated Fatty Liver Disease (MAFLD), with new and easy diagnostic criteria need to be discussed. In particular, it will be nice to underline how the NAFLD diagnosis have been established with respect the use of alcohol, and if possible to evaluate how many of the NAFLD diagnosed patients also respect the MAFLD diagnosis.

In our study NAFLD are patients in the OPTN database recorded as having a diagnosis of cirrhosis from fatty liver (NASH, non-alcoholic steatohepatitis) (Table S1 in the supplementary supporting information). 

Metabolic dysfunction-associated fatty liver disease (MAFLD), the newly proposed overarching terminology for fatty liver disease introduced in 2020, is based on the presence of hepatic steatosis and any 1 of 3 metabolic risks, namely obesity (BMI >25), DM Type 2, or metabolic dysregulation (2 of following: high waist circumference, high blood pressure, high cholesterol, pre-diabetes, insulin resistance, and high plasma C-reactive protein levels)

The OPTN database has data on hepatic steatosis entered as NASH, as well as BMI and DM Type 2 status, but does not capture data on metabolic dysregulation. Based on the available data reported in the OPTN database (diagnosis of NASH, and either BMI >25 or presence of DM Type 2) an estimated 94.3 to 97.3% of NAFLD transplants each year would satisfy the MAFLD criteria.

NAFLD transplants satisfying MAFLD criteria

Year 2010 2011 2012 2013 2014 2015 2016 2017 2018 2019

Satisfies criteria 496

(95.6%) 515

(96.4%) 579

(97.3%) 662

(94.8%) 747

(94.3%) 864

(94.7%) 1200

(94.9%) 1308

(95.5%) 1475

(95.5%) 1705

(95.0%)

Total 519 534 595 698 792 912 1264 1370 1544 1795

3) Since this topic is very debated, I would like to see and compare the results of the papers facing the same topic of the changing etiology of transplanted HCC.

There have been no reports on the etiology of HCC among patients who received liver transplant after 2015 in the US, and no comprehensive study on all causes of HCC among liver transplant recipients. 

We added a sentence in the discussion that our findings in the trends in etiology of HCC among liver transplant recipients are consistent with the recent study by Wong and Singal who reported an increasing trend in liver transplant waitlist registrants with HCC associated with NASH and ALD and the decreasing trend associated with HCV. In the first 3 months of 2019, among women with HCC on the liver transplant waitlist, NASH has surpassed HCV (including HCV+ALD), but HCV remained the major etiology among men on the liver transplant waitlist with HCC. (Wong RJ, Singal AK. Trends in Liver Disease Etiology Among Adults Awaiting Liver Transplantation in the United States, 2014-2019. JAMA Network Open. 2020;3(2):e1920294. doi:10.1001/jamanetworkopen.2019.20294)

4) The diagnosis of NAFLD, ALD, cryptogenic, and metabolic liver cirrhosis should be carefully and precisely defined.

The diagnoses of NAFLD, ALD, cryptogenic, and metabolic liver cirrhosis were defined according to the OPTN diagnoses as listed in Table S1 in the supplementary supporting information: 

NAFLD: NASH diagnosis 

ALD: alcoholic cirrhosis without HCV, acute alcoholic hepatitis

Cryptogenic: cryptogenic cirrhosis WITHOUT other secondary diagnosis

Metabolic: Cystic fibrosis, A1AD, Wilson’s disease, hemochromatosis, glycogen storage disease, hyperlipidemia, tyrosinemia, oxaluria, maple syrup urine disease, other metabolic disease

Accuracy of the OPTN database is dependent on the diagnoses and data entries by the physicians associated with each transplant center.

5) It should be mentioned whether other risk factors of NAFDL/MAFLD are known, e.g., DM II, metabolic syndrome, etc...

The OPTN database has DM II and BMI data, but we do not have data on triglyceride levels, metabolic syndrome, PCOS, thyroid function, or pituitary function. The lack of information in the OPTN database on metabolic syndrome and risk factors of NAFLD was added under the limitations of the study. 

6) Many patients with cryptogenic liver cirrhosis had in their clinical history previous diagnosis of metabolic risk factors, which may also disappear as the liver disease evolves. When in the anamnesis we retrieve these factors, even if not present more, we prefer to attribute the etiology to NAFLD rather than to unknown causes. Please discuss whether anamnestic history can be considered to evaluate the etiologies.

Unfortunately, the OPTN STAR database does not include variables for metabolic risk factors such as a large waistline, high triglyceride level, low HDL cholesterol level, high blood pressure, or high fasting blood sugar. We also cannot access patients’ history aside from the diagnoses recorded on the intake form or at time of transplant.

---

## [Decision Letter · Decision Letter 1]

8 Sep 2020

Causes and trends in liver disease and hepatocellular carcinoma among men and women who received liver transplants in the U.S., 2010-2019.

PONE-D-20-21329R1

Dear Dr. Samuel K So,

We’re pleased to inform you that your manuscript has been judged scientifically suitable for publication and will be formally accepted for publication once it meets all outstanding technical requirements.

Kind regards,

Gianfranco D. Alpini

Academic Editor

PLOS ONE

Additional Editor Comments (optional):

Reviewers' comments:

Reviewer's Responses to Questions

**Comments to the Author**

1. If the authors have adequately addressed your comments raised in a previous round of review and you feel that this manuscript is now acceptable for publication, you may indicate that here to bypass the “Comments to the Author” section, enter your conflict of interest statement in the “Confidential to Editor” section, and submit your "Accept" recommendation.

Reviewer #1: All comments have been addressed

Reviewer #2: All comments have been addressed

2. Is the manuscript technically sound, and do the data support the conclusions?

Reviewer #1: (No Response)

Reviewer #2: (No Response)

3. Has the statistical analysis been performed appropriately and rigorously? 

Reviewer #1: (No Response)

Reviewer #2: (No Response)

4. Have the authors made all data underlying the findings in their manuscript fully available?

Reviewer #1: (No Response)

Reviewer #2: (No Response)

5. Is the manuscript presented in an intelligible fashion and written in standard English?

Reviewer #1: (No Response)

Reviewer #2: (No Response)

6. Review Comments to the Author

Reviewer #1: (No Response)

Reviewer #2: All comments have been nicely addressed in this version. The manuscript seems ameliorated significantly

7. PLOS authors have the option to publish the peer review history of their article (what does this mean?). If published, this will include your full peer review and any attached files.

Reviewer #1: No

Reviewer #2: No

---

## [Editor Report · Acceptance letter]

10 Sep 2020

PONE-D-20-21329R1 

Causes and trends in liver disease and hepatocellular carcinoma among men and women who received liver transplants in the U.S., 2010-2019. 

Dear Dr. So:

I'm pleased to inform you that your manuscript has been deemed suitable for publication in PLOS ONE. Congratulations! Your manuscript is now with our production department. 

Kind regards, 

on behalf of

Dr. Gianfranco D. Alpini 

Academic Editor

PLOS ONE